# Morphological and physiological characterization of filamentous *Lentzea aerocolonigenes*: Comparison of biopellets by microscopy and flow cytometry

Kathrin Schrinner[1,2], Lukas Veiter[3,4], Stefan Schmideder[5], Philipp Doppler[3], Marcel Schrader[2,6], Nadine Münch[5], Kristin Althof[1], Arno Kwade[2,6], Heiko Briesen[5], Christoph Herwig[3], Rainer Krull[1,2]*

**1** Institute of Biochemical Engineering, Technische Universität Braunschweig, Braunschweig, Germany,
**2** Technische Universität Braunschweig, Center of Pharmaceutical Engineering, Braunschweig, Germany,
**3** Institute of Chemical, Environmental and Bioscience Engineering, Research Area Biochemical Engineering, Technische Universität Wien, Vienna, Austria, **4** Competence Center CHASE GmbH, Linz, Austria, **5** School of Life Sciences, Chair of Process Systems Engineering, Technische Universität München, Freising, Germany, **6** Institute for Particle Technology, Technische Universität Braunschweig, Braunschweig, Germany

* r.krull@tu-braunschweig.de

**Data Availability Statement:** All relevant data are within the paper and its Supporting Information files.

**Funding:** The authors (KS, SS, MS, AK, HB, RK) gratefully acknowledge financial support by the German Research Foundation (DFG) in the Priority Programme 1934 DiSPBiotech – Dispersity, structural and phase modifications of proteins and biological agglomerates in biotechnological processes (SPP 1934 DiSPBiotech – 315384307

## Abstract

Cell morphology of filamentous microorganisms is highly interesting during cultivations as it is often linked to productivity and can be influenced by process conditions. Hence, the characterization of cell morphology is of major importance to improve the understanding of industrial processes with filamentous microorganisms. For this purpose, reliable and robust methods are necessary. In this study, pellet morphology and physiology of the rebeccamycin producing filamentous actinomycete *Lentzea aerocolonigenes* were investigated by microscopy and flow cytometry. Both methods were compared regarding their applicability. To achieve different morphologies, a cultivation with glass bead addition (Ø = 969 μm, 100 g L$^{-1}$) was compared to an unsupplemented cultivation. This led to two different macro-morphologies. Furthermore, glass bead addition increased rebeccamycin titers after 10 days of cultivation (95 mg L$^{-1}$ with glass beads, 38 mg L$^{-1}$ without glass beads). Macro-morphology and viability were investigated through microscopy and flow cytometry. For viability assessment fluorescent staining was used additionally. Smaller, more regular pellets were found for glass bead addition. Pellet diameters resulting from microscopy followed by image analysis were 172 μm without and 106 μm with glass beads, diameters from flow cytometry were 170 and 100 μm, respectively. These results show excellent agreement of both methods, each considering several thousand pellets. Furthermore, the pellet viability obtained from both methods suggested an enhanced metabolic activity in glass bead treated pellets during the exponential production phase. However, total viability values differ for flow cytometry (0.32 without and 0.41 with glass beads) and confocal laser scanning microscopy of single stained pellet slices (life ratio in production phase of 0.10 without and 0.22 with glass beads), which is probably caused by the different numbers of investigated pellets. In confocal laser scanning microscopy only one pellet per sample could be investigated while flow

(HB) and 315457657 (RK/AK)). The work of LV and CH was supported by the Austrian Research Promotion Agency (FFG) (Grant number: 844608) and within the framework of the Competence Center CHASE GmbH, funded by the Austrian Research Promotion Agency (grant number 868615) as part of the COMET program-Competence Centers for Excellent Technologies by BMVIT, BMDW, the Federal Provinces of Upper Austria and Vienna. The funders provided support in the form of salaries for authors (KS, LV, SS, MS), but did not have any additional role in the study design, data collection and analysis, decision to publish, or preparation of the manuscript. The specific roles of these authors are articulated in the 'author contributions' section. We also acknowledge support by the Open Access Publication Funds of the Technische Universität Braunschweig.

**Competing interests:** Competence Center CHASE GmbH participated in the salary of LV. This does not alter our adherence to PLOS ONE policies on sharing data and materials.

cytometry considered at least 50 pellets per sample, resulting in an increased statistical reliability.

## Introduction

One of the most sensitive process characteristics in the cultivation of filamentous biological systems is their complex morphology [1]. It is well known that there is a strong relationship between the cultivation process conditions, the cell morphology, and the productivity of filamentous microorganisms [2–5]. Thus, the cell morphology is a key aspect in understanding filamentous microorganisms. The micro-morphology of filamentous microorganisms consists of cylindrical fibers called *hyphae*, which grow and branch. During submerged cultivations, the macro-morphology varies–depending on the cultivation conditions–between freely dispersed mycelium to pellets, densely packed filamentous agglomerates [1, 6]. The development of the macro-morphology is mainly driven by aggregation, fragmentation, and growth [7]. Dense hyphal networks are known to limit the transport of substrates into filamentous pellets and thus also influence their viability and productivity [8–10].

According to Papagianni [11], light microscopy and subsequent (semi-)automated image analysis (e.g., [12, 13]) are the state-of-the-art method to investigate the macro-morphology of filamentous microorganisms. However, microscopy can provide information about the outer structure on the one hand and on the other hand about the inner structure of pellets by slicing them first [14]. Since manual pellet slicing takes a lot of time and precise manual work, the process is not sufficiently validated statistically. So far, the only non-destructive method that is capable to analyze the actual 3D micro-morphology, i.e., the hyphal network, within whole filamentous pellets is based on X-ray microcomputed tomography [15]. Nowadays, this method is limited to filamentous microorganisms with hyphal diameters $> 3$ μm and is thus not suitable for most filamentous bacteria (e.g., *Lentzea aerocolonigenes* with 0.5 μm hyphal diameter [16]). Even more comprehensive understanding of the morphology's impact on substrate transport and viability in pellets can be obtained through modeling or experimental approaches. To model the transport of substrates into pellets, their detailed micro-morphology has to be known, which was shown for filamentous fungi (*Aspergillus niger*) with hyphal diameters $> 3$ μm [17]. Experimentally, a combination of viability staining and confocal laser-scanning microscopy (CLSM) facilitates the distinction between active and inactive regions inside individual pellets off-line [10]. However, to achieve in-process monitoring of the viability, a quantitative approach is necessary. A quantitative determination of the viability can be achieved with at-line chemical methods such as fluorescent staining or physical techniques using various sensors like dielectric spectroscopy, infrared spectroscopy, and fluorescence [18, 19]. While these methods enable real time measurement, morphological aspects are ignored. Contrary, flow cytometry enables a combined assessment of cell morphology and cell physiology via scatter light and the use of fluorescent viability staining. Further benefits encompass a high statistical robustness, low measurement times, and potential on-line applicability [9].

The applied microorganism in this study, *Lentzea aerocolonigenes*, is a filamentous actinobacterium which produces the secondary metabolite rebeccamycin, among others [16, 20]. Rebeccamycin gained interest because of its antibiotic and antitumor properties [16, 21]. An analogue, called becatecarin, with an improved water-solubility has been promising in clinical trials (phase I and II) for the treatment of refractory breast cancer, metastatic colorectal cancer

and small cell lung cancer [22–24]. However, research on the relation between productivity, morphology and cultivation process conditions for *L. aerocolonigenes* is scarce.

One promising way to produce a favourable macro-morphology is the addition of biologically and chemically inert particles. Microparticle (Ø < 50 μm) addition to cultivations of filamentous bacteria or fungi often showed positive effects regarding productivity and is a frequently used approach in recent years [25–29]. A less frequently utilized approach is the addition of larger (glass) particles (Ø > 250 μm) [30, 31]. This has already been successfully applied for *L. aerocolonigenes* leading to a significant increase in rebeccamycin concentration [29, 32].

The aim of this study was the macro-morphological and physiological investigation of *L. aerocolonigenes* cultivations by applying and comparing two different morphology-determining methods. For this purpose, a glass bead supplemented and an unsupplemented cultivation were used to generate different cell morphologies. Cell morphology and viability of the microorganism were examined and compared using microscopy and flow cytometry combined with further analytical methods. Both methods, microscopy and flow cytometry, enabled the analysis of several thousand pellets.

## Material and methods

### Cultivation

Filamentous *Lentzea aerocolonigenes* (DSM 44217), purchased from the German Collection of Microorganisms and Cell Cultures (DSMZ, Braunschweig, Germany), was cultivated for investigations in this work. A pre-culture was inoculated with 1 mL of frozen culture (sample of a two day pre-culture frozen at -80˚C in 30% glycerol). A 250 mL baffled shaking flask with 50 mL GYM medium (4 g L$^{-1}$ glucose, 4 g L$^{-1}$ yeast extract, 10 g L$^{-1}$ malt extract) were incubated for two days on an orbital shaker (Certomat BS-1, Sartorius, Göttingen, Germany) with a shaking diameter of 50 mm and a shaking frequency of 120 min$^{-1}$ at 28˚C in darkness. The pH of the GYM medium was adjusted to 7.2 via 2 M KOH before sterilization, glucose was sterilized separately. The inoculation of the main-culture was conducted by transferring 300 μL of the pre-culture in each shaking flask. Baffled shaking flasks with a total volume of 250 mL filled with 50 mL GYM medium were used. The main-cultures were incubated for 10 days as described for the pre-culture and sampled daily. In some cases, glass beads (100 g L$^{-1}$, mean diameter of 969 μm, Sigmund Lindner, Warmensteinach, Germany) were added to the cultivation medium to test potential effects on pellet morphology and productivity.

### Rebeccamycin quantification

Extraction of rebeccamycin from 20 mL cultivation broth was performed using 5 mL ethyl acetate which was incubated for 60 min in an overhead shaker (Intelli-Mixer RM-2 M, LTF Labortechnik, Wasserburg, Germany). After centrifugation for 10 min at 4000 min$^{-1}$ (Heraeus Varifuge 3.0R, Thermo Fisher Scientific, Waltham, USA) the ethyl acetate was removed and used for HPLC measurements (HitachiLaChrom Elite, Hitachi, Tokyo, Japan). For HPLC analysis a pre-column Hypersil ODS (5 μm, 50 × 4.6 mm, Thermo Fisher Scientific, Waltham, USA) and a column Hypersil ODS (5 μm, 250 × 4.6 mm, Thermo Fisher Scientific, Waltham, USA) at 30˚C were used. A gradient of two eluents, 0.1% trifluoric acid (eluent A) and 90% acetonitrile (eluent B), was used at a flow rate of 1 mL min$^{-1}$. Starting at 77.8% of eluent A, it was reduced to 16.6% eluent A over 20 min, kept constant for 2 min and was increased to 77.8% again. A diode array detector (Hitachi L-2455 Diode Array Detector, Hitachi, Tokyo, Japan) detected rebeccamycin at a wavelength of 316 nm with a retention time of approximately 18 min.

The biomass-specific rebeccamycin productivity is calculated by

$$q_P \left[ \mathrm{mg\ g^{-1} d^{-1}} \right] = \frac{1}{X} \cdot \frac{dP}{dt} \qquad (1)$$

with the cell dry weight concentration X and the rebeccamycin concentration P.

## Glucose quantification

Glucose concentration was determined via HPLC (HitachiLaChrom Elite, Hitachi, Tokyo, Japan) using water as eluent at a flow rate of 0.6 mL min$^{-1}$. For analysis, the pre-column Meta-carb 87C Guard (50 × 4.6 mm, Agilent Technologies, Santa Clara, USA) and the column Meta-carb 87C (300 × 7.8 mm, Agilent Technologies, Santa Clara, USA) at 85°C followed by an RI detector (Hitachi L-2490 RI detector, Hitachi, Tokyo, Japan) were applied with a retention time of approximately 11 min.

## Cell dry weight concentration determination

Cell dry weight concentration of the biomass was determined gravimetrically. Filter papers (Quantitative Papers/Grade 389, Sartorius, Göttingen, Germany) were dried and pre-weighed. 5 mL of the samples were filtered and subsequently dried at 105°C for 48 h. Cell dry weight determination was conducted in duplicates.

## Microscopic morphological analysis

To investigate the macro-morphology, 40 microscopic pictures of each sample were taken with a digital inverse microscope (EVOS XL, AMG, Bothell, WA, USA). Therefore, 300 μL cultivation broth and 7 mL demineralized water were mixed on a petri dish (94 × 16 mm, polystyrene, Greiner Bio-One, Kremsmünster, Austria), that was marked for 40 non-overlapping microscopic pictures (area of 9.2 × 6.9 mm). The applied resolution for all microscopic pictures was 4.5 μm per pixel. Image analysis was conducted using Matlab (version 2019a, Math-Works, Natick, MA). First, the images were binarized with *adaptthresh*, an adaptive image threshold using local mean intensities in the neighbourhood of each pixel. To segment pellets, watershed segmentation was applied. Afterwards, morphological properties of the segmented objects such as projected area, perimeter, circularity, and Feret diameter were determined with the Matlab function *regionprops*. Finally, small objects and objects touching the edge of the image were deleted automatically. Small objects were defined as objects with a projected area < 200 μm$^2$. The area-equivalent-spherical-diameter ($d_a$) and circularity were defined as:

$$d_a \, [\mathrm{m}] = \sqrt{\frac{4 \cdot Area}{\pi}} \qquad (2)$$

$$Circularity \, [-] = 4\pi \cdot \frac{Area}{Perimeter^2} \qquad (3)$$

## Viability staining, pellet slicing and imaging with CLSM

The pellets of *L. aerocolonigenes* were stained with SYTO9 (Molecular Probes, Eugene, USA) and propidium iodide (PI) (Molecular Probes, Eugene, USA). To prepare the staining solution 1.5 μL 3.34 mM SYTO9 in DMSO and 1.5 μL 20 mM PI in DMSO were added to 1 mL 1x salt solution (0.31 g L$^{-1}$ ammonium chloride, 4.33 g L$^{-1}$ disodium hydrogen phosphate, 0.13 g L$^{-1}$ potassium chloride, 3.04 g L$^{-1}$ sodium dihydrogen phosphate dihydrate) and mixed thoroughly. Individual pellets of the cultivation broth were transferred into the staining solution

and incubated for 15 min at room temperature in darkness. Afterwards, the pellets were frozen at -20˚C in liquid cryosectioning medium (Neg-50, Richard Allan Scientific, USA). These frozen pellets were sliced by a cryomicrotome (HM550, Microm, Neuss, Germany). Each stained pellet frozen in cryosectioning medium was fixed with liquid cryosectioning medium on the rapid freezing station of the cryomicrotome and was frozen again. The pellets were sliced in sections of 100 μm thickness and a section from the equatorial region was transferred onto an object slide.

Pellet sections were monitored using a CLSM (C2si, Nikon Instruments, Amsterdam, The Netherlands) with a 4x objective. The living pellet areas were observed with a 510–540 nm laser and the dead areas with a 620–650 nm laser, resulting in a green appearance of living areas and a red appearance of dead areas. Red fluorescence is caused by PI which is intercalating with nucleic acid, but is impermeable to cell membranes. SYTO9 is also intercalating with nucleic acids, however, it can permeate membranes leading to a green fluorescence of all cells containing nucleic acid. PI has a stronger affinity for nucleic acid so that PI displaces SYTO9 when both stains are present [33]. Separate images for green and red areas were acquired (see **Fig 1** for raw and processed CLSM images). Image analysis was conducted using Matlab (version 2019a, MathWorks, Natick, MA). First, the green and red images were binarized applying Otsu's method [34] on the green and red channels, respectively. Holes in the binarized images were filled using *imclose* with a structuring element of 5 pixels. By the intersection of the binarized red and green images, the pellet sections could be differed into three regions: solely green/living areas, solely red/dead areas, and areas containing both, living and dead areas. From these regions, the share of each area in the total area was calculated. The live ratio was defined as:

$$Live\ ratio\ [-] = \frac{Living\ area}{Total\ area} \qquad (4)$$

## Flow cytometry

Samples from cultivations were diluted 1:10 into phosphate buffered saline (50 g L$^{-1}$ of 2.65 g L$^{-1}$ CaCl$_2$ solution, 0.2 g L$^{-1}$ KCl, 0.2 g L$^{-1}$ KH$_2$PO$_4$, 0.1 g L$^{-1}$ MgCl $\cdot$ 6 H$_2$O, 8 g L$^{-1}$ NaCl and 0.764 g L$^{-1}$ Na$_2$HPO$_4$ + 2 H$_2$O) and stained with PI (Sigma Aldrich, St. Louis, Missouri/USA; 20 mM stock dissolved in DMSO ≥ 99.9%, diluted with phosphate buffered saline to a final concentration of 20 μM). After incubating 1 min, the sample was further stained with fluorescein diacetate (Sigma Aldrich, St. Louis, Missouri, USA; stock solution of 5 g L$^{-1}$ dissolved in acetone ≥ 99.9% to a final concentration of 5 mg L$^{-1}$). After an incubation time of 5 min, the sample was further diluted (1:100 in the same buffer) for flow cytometric analysis. Metabolic activity is shown by fluorescein diacetate (FDA) treatment resulting in green fluorescence through esterase activity. Red fluorescence from PI is a result from DNA intercalation in cells with compromised membranes [35]. **S1 Fig** in the supporting information displays an exemplary flow cytometry scatterplot depicting populations of viable biomass (green), non-viable biomass (red) and background (brown) based on green (FDA positive) and red fluorescence (PI positive) signals.

A CytoSense flow cytometer (CytoBuoy, Woerden, The Netherlands) was used for all measurements as described previously [9, 35]. Measurements were performed at a sideward scatter (SSC) trigger level of 200 mV based on the expected size of biomass particles as a threshold for data acquisition. Measurement duration was kept at 90 s employing a flow rate of the sample pump at 3 μl per second, a total number of 3000–8000 pellets were registered in each measurement process before data analysis. The measurement software CytoUSB (CytoBuoy, Woerden, The Netherlands) does not allow compensation of fluorescence spectral overlap which might arise from simultaneous use of more than one fluorescent dye. However, this issue is dealt with efficiently by regularly checking the ratio between red and green fluorescence. Deviations

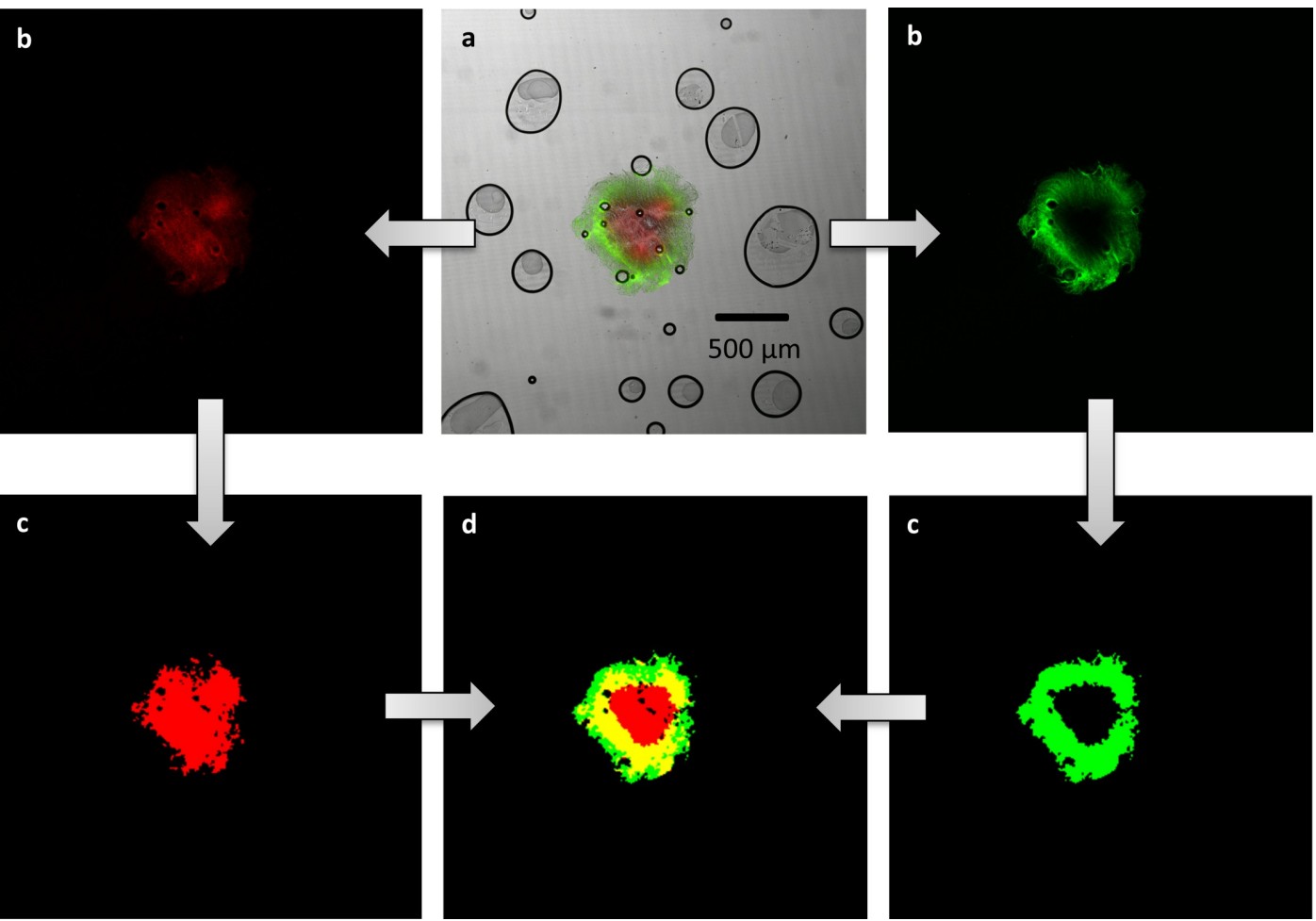

**Fig 1.** Processing scheme for the image analysis of stained pellet slices: (a) original CLSM image with green and red fluorescent areas (black circles are air bubbles that are occasionally enclosed in the sectioning medium), (b) separation in green fluorescence and red fluorescence, (c) images after binarization and closing and (d) combination to one image with living (green), dead (red) or living and dead pellet areas (yellow).

in this ratio occur due to spectral overlap and corresponding data can be corrected or omitted [9, 35]. Data analysis was performed using the software CytoClus4 (CytoBuoy, Woerden, The Netherlands). The CytoSense flow cytometer provides multiple data points per channel per pellet. This signal shape is achieved for both scatter channels as well as green (FLG) and red fluorescence (FLR) channels [36]. These pulse shapes are the basis for multiple curve parameters [37]. This method enables classification of cell morphology according to size and form based on forward scatter (FSC) in combination with sideward scatter (SSC): small elements, large elements and pellets. Pellets are distinguished by a saturation of FSC signal in the particle core and an overall signal length > 80 μm.

For pellets, the parameter "compactness" representing the density of the pellet can be derived from the analysis of SSC signal length in combination with pellet size, hereafter termed "Compactness according to SSC" and calculated according to:

$$Compactness_{SSC} \, [-] = \frac{Length \; of \; SSC \; signal \; [\mu m]}{Signal \; length \; [\mu m]} \tag{5}$$

"Length of SSC signal" is based on the signal length at half of the maximum of the signal level of the signal profile. To compensate for increased signals at high pellet sizes, "Length of SSC signal" is divided by "Signal length" which represents the overall diameter of the pellet. It should be noted that size exclusion effects due to the diameter of the sampling tube are possible. This excludes particles at diameters above 500 µm and hinders measurement of particles at diameters above 300 µm.

Assessment of pellet viability is based on individual pellet signal shapes. In order to estimate contributions of green fluorescence signals from FDA staining to viability, viability is calculated according to:

$$Viable\ layer\ vl_{FDA}\ [\mu m] = \frac{Area\ under\ FLG\ curve\ [mV \cdot \mu m]}{Area\ under\ FSC\ curve [mV \cdot \mu m]} \cdot signal\ length\ [\mu m] \cdot 0.5 \quad (6)$$

If red fluorescence signals from PI staining are to be considered, the viable layer is calculated based on the individual maximum of fluorescence signals. In Eq (7) this maximum is considered as a "threshold" value defined as 0.3 of the maximum fluorescence.

$$Viable\ layer\ vl_{PI}\ [\mu m] = 0.5 \cdot (1 - Length\ of\ FLR > threshold\ [\mu m]) \quad (7)$$

In Eq (6) and Eq (7) the factor 0.5 is included to obtain the viable layer as a simplified radius since the pellet is considered as a sphere.

To further estimate pellet viability and demonstrate the relation of viable layer according to Eq (6) and Eq (7) to pellet size, a viability factor was calculated according to:

$$Viability\ factor\ [-] = \frac{2 \cdot viable\ layer\ [\mu m]}{pellet\ size\ [\mu m]} \quad (8)$$

To assess the autofluorescence of the biomass, several measurements were conducted without the use of fluorescent staining. Assessment of autofluorescence was based on the calculation of the autofluorescence factor according to:

$$Autofluorescence\ factor\ [-] = \frac{Area\ under\ FLG\ curve\ [mV \cdot \mu m]}{Area\ under\ FSC\ curve\ [mV \cdot \mu m]} \quad (9)$$

Detailed information on the flow cytometry method can be found in [9].

## Statistical analysis

Single-factor analysis of variance (ANOVA) was conducted for the statistical analysis of the results from this study. The statistical test was done in Microsoft Excel with $\alpha = 0.05$. A p-value lower than 0.05 (or an F-value larger than the critical F-value) indicates a statistically significant difference between two means. The results with and without glass bead addition were compared for each day of cultivation. Results with very low numbers of samples were not considered for statistical analysis as type II errors are prone to occur.

## Results

### Rebeccamycin formation with glass bead addition

The positive effects of glass bead addition on rebeccamycin formation by *L. aerocolonigenes* have already been mentioned in literature [29, 32]. Cultivation conditions were similar as described by Schrader et al. [32] where the effect of different glass bead diameters was investigated. Based on that study, a more detailed investigation of glass bead effects by comparing a cultivation with 969 µm mean diameter glass beads with a glass bead concentration of 100 g L$^{-1}$

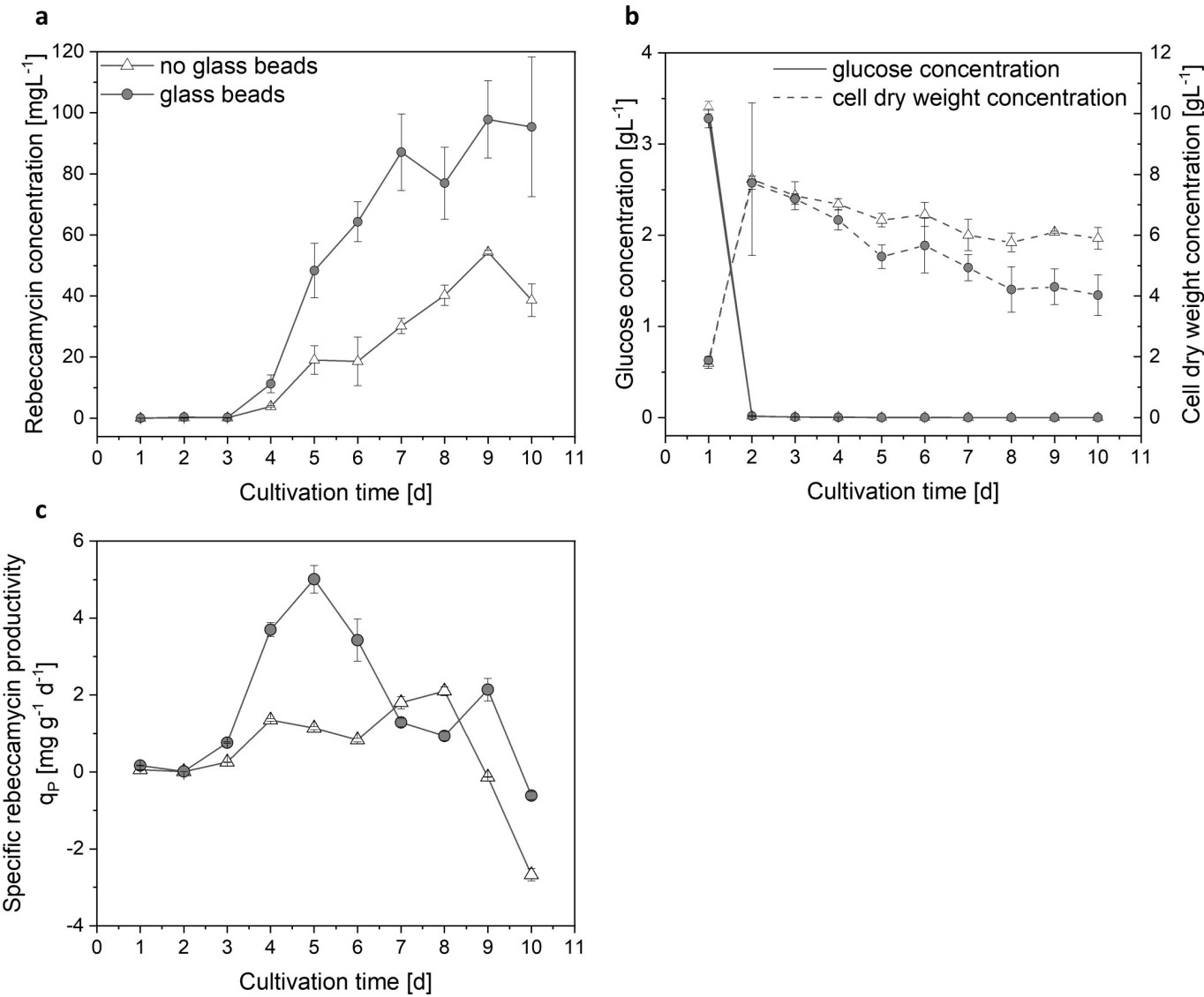

**Fig 2.** (a) Rebeccamycin, (b) glucose and cell dry weight concentrations and (c) specific rebeccamycin productivity $q_P$ of a 10 day cultivation of *L. aerocolonigenes* <u>with</u> (circle) and <u>without</u> (triangle) glass bead addition (Ø = 969 μm, 100 g L$^{-1}$). Presented data are mean values with their respective standard deviations.

to an unsupplemented cultivation was conducted in this study. **Fig 2** illustrates the resulting rebeccamycin, glucose, and cell dry weight concentrations as well as the specific rebeccamycin productivity $q_P$ for this comparison. Between day 3 and 4 of cultivation the rebeccamycin production begins for both approaches (**Fig 2A**). The rebeccamycin concentration for the glass bead supplemented cultivation increases faster than for the unsupplemented cultivation resulting in a higher final concentration of about 95 mg L$^{-1}$ after 10 days compared to around 38 mg L$^{-1}$ without glass beads. Vigorous biomass growth is observed until day 2 for both approaches (**Fig 2B**). The cell dry weight concentrations display the same maximum but are then decreasing slightly faster with glass bead addition. The glucose consumption and growth phases for both approaches are quite similar indicating that the glass bead addition does not directly affect growth kinetics. It is neither slowing down nor accelerating growth. The specific rebeccamycin

productivity $q_P$ (Fig 2C) is significantly increased with glass bead addition, especially in the exponential production phase. The maximal $q_P$ at day 5 of cultivation is approximately 4-fold higher with glass bead addition compared to the unsupplemented cultivation.

In order to determine the impact of cell morphological aspects on productivity, several analytical methods were employed which will be discussed in the following sections.

### Macro-morphological assessment via microscopic image analysis

The effect of glass bead addition on the macro-morphology was investigated through microscopic measurements and subsequent automated image analysis. To get a statistical insight into macro-morphological properties such as area, perimeter, area-equivalent-spherical-diameter, circularity, and Feret diameter, 40 pictures per cultivation per day were analyzed resulting in approximate 158.600 pellets. 54.800 pellets originated from cultivations without glass beads, whereas 103.800 pellets were analyzed from cultivations with glass beads. The number-density-distributions $q_0(d_a)$ (Fig 3A) of all pellets analyzed during the 10 days of cultivation shows that the cultivations with glass beads results in smaller pellets than the cultivations without

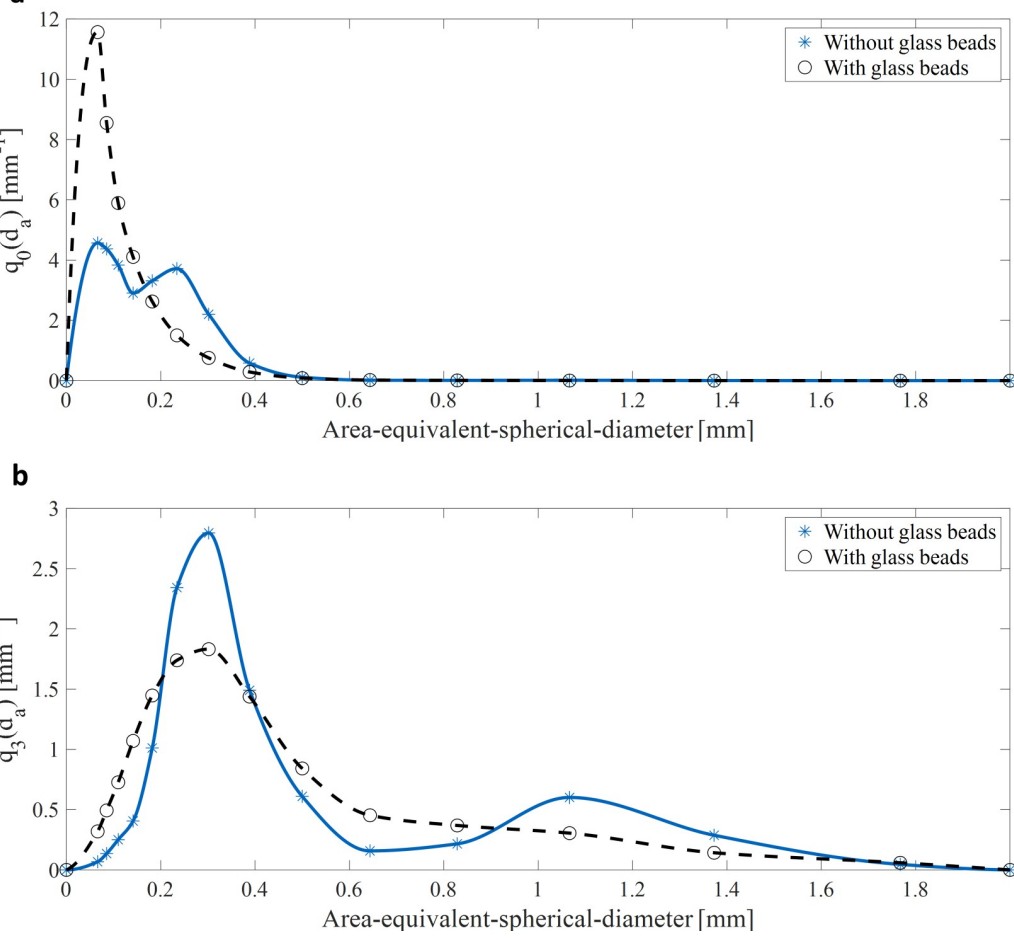

**Fig 3.** (a) Normalized number-density-distribution $q_0(d_a)$ and (b) volume-density-distribution $q_3(d_a)$ of all pellets analyzed through microscopic image analysis during the 10 days of cultivation ($d_a$, mean area-equivalent-spherical-diameter of the pellet (Eq (2)). Blue star-shaped data points (n = 54.800) represent the cultures processed <u>without</u> glass beads, whereas black circular data points (n = 103.800) show cultures processed <u>with</u> glass beads.

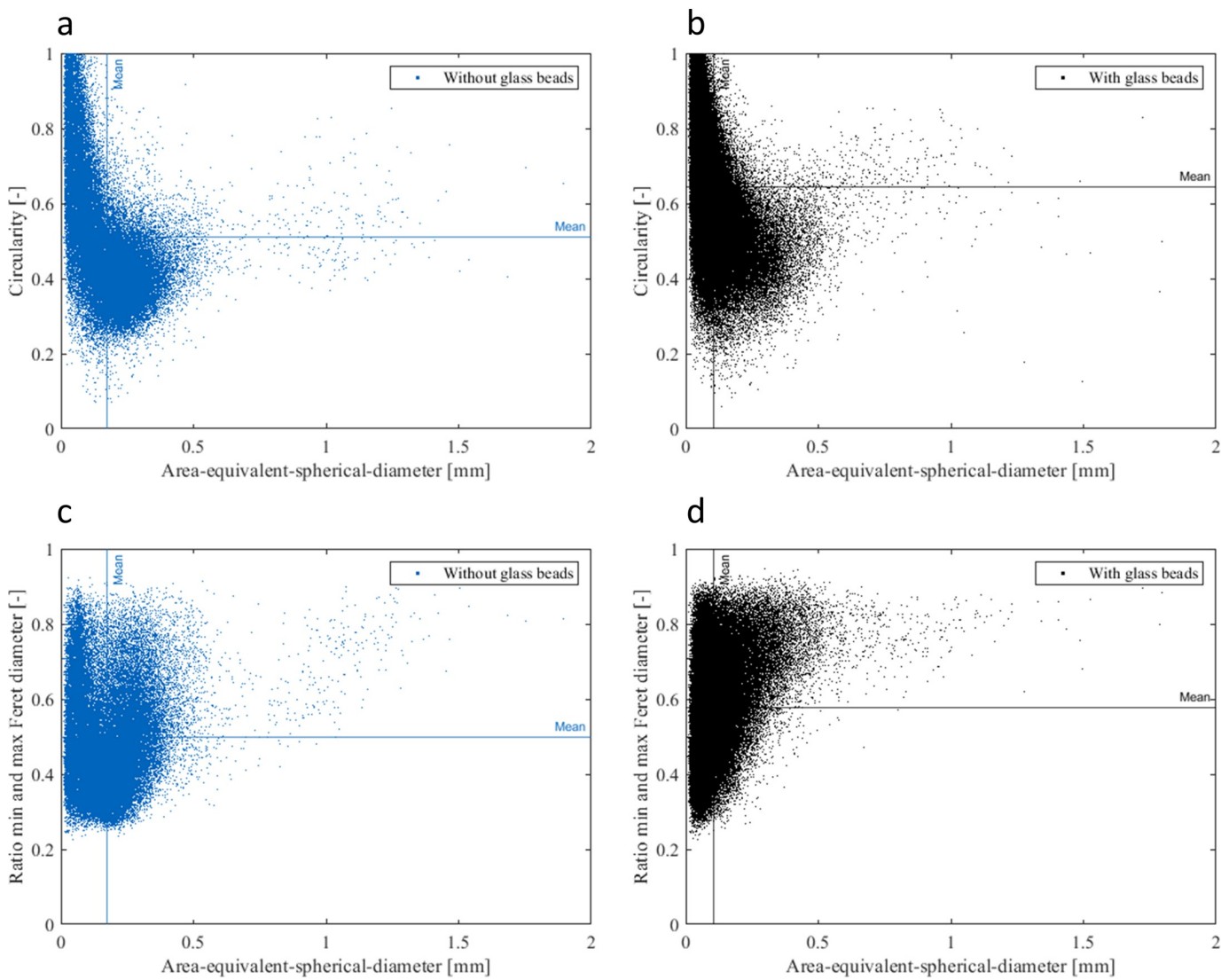

**Fig 4. Morphological properties of all pellets analyzed through microscopic image analysis during the 10 days of cultivation.** Blue data points (n = 54.800) represent the cultures processed underline{without} glass beads, whereas black data points (n = 103.800) show cultures processed underline{with} glass beads. Circularity (Eq (3)) and area-equivalent-spherical-diameter (a) underline{without} and (b) underline{with} glass beads. Ratio between minimum and maximum Feret diameter and area-equivalent-spherical-diameter (c) underline{without} and (d) underline{with} glass beads.

glass beads. The mean area-equivalent-spherical-diameter of the pellet ($d_a$, Eq (2)) of the cultivations with and without glass beads was 106 and 172 μm, respectively. In general, $d_a$ of almost all pellets of both cultivations ranges between 0 and 500 μm. Most of the volume of the pellets can be also found in this size-range (**Fig 3B**). However, due to the high volume of the few bigger pellets, their volume fraction is not negligible. The number- and volume-density distributions of both cultivations (**S2 Fig**) do not change drastically during the production of rebeccamycin (day 3–9 of cultivation).

Besides the area-equivalent-spherical-diameter, the circularity (Eq (3)) and the ratio between the minimum and maximum Feret diameter have also been investigated (**Fig 4**). The mean of the circularity of all pellets from the cultivations processed without glass beads is 0.51 ± 0.09 (**Fig 4A**) and processed with glass beads is 0.64 ± 0.09 (**Fig 4B**), respectively. The

mean of the ratio between the minimum to the maximum Feret diameter, also a measure for the roundness of pellets, of the cultivations processed without glass beads with a value of 0.50 ± 0.07 (**Fig 4C**) is also lower than for the cultivations processed with glass beads with 0.58 ± 0.07 (**Fig 4D**).

In summary, pellets of the cultivation treated with glass beads were slightly smaller and more circular than pellets of the cultivation processed without glass beads.

## Viability assessment via CLSM

To further shed light on parameters affecting rebeccamycin production, the viability of pellets was investigated using additional analytical methods. To identify metabolically active zones in the heterogeneous pellet structure, CLSM measurements and subsequent image analysis of live/dead stained pellet-slices were conducted (compare **Fig 1**). The share of live (green color), dead (red color), and overlapping areas (yellow color) of the pellets is presented in **Fig 5**. Due to the high manual effort of pellet slicing only one pellet per sample point was examined. Despite this small pellet number resulting in a low statistical reliability, this method was used for a qualitative assessment. By comparing the live area of the pellets without (**Fig 5A**) and with glass beads (**Fig 5B**), higher values with glass bead addition can be seen between 3 to 8 days cultivation time. During this time a large increase in the rebeccamycin concentration (**Fig 2A**) is visible proposing that a larger proportion of living cell material leads to a higher product formation. During the same cultivation period, the share of dead pellet biomass (red) increases slightly, while the proportion of living/dead biomass (overlapping area, yellow) decreases.

## Morphological and physiological assessment via flow cytometry

In order to increase statistical reliability a larger number of pellets was characterized by flow cytometry. The here-presented flow cytometry method allows for individual pellet analysis via spatially resolved signal profiles which represent approximations of the actual pellet's cross section as previously described in [9]. **Fig 6** demonstrates these profiles for individual pellets

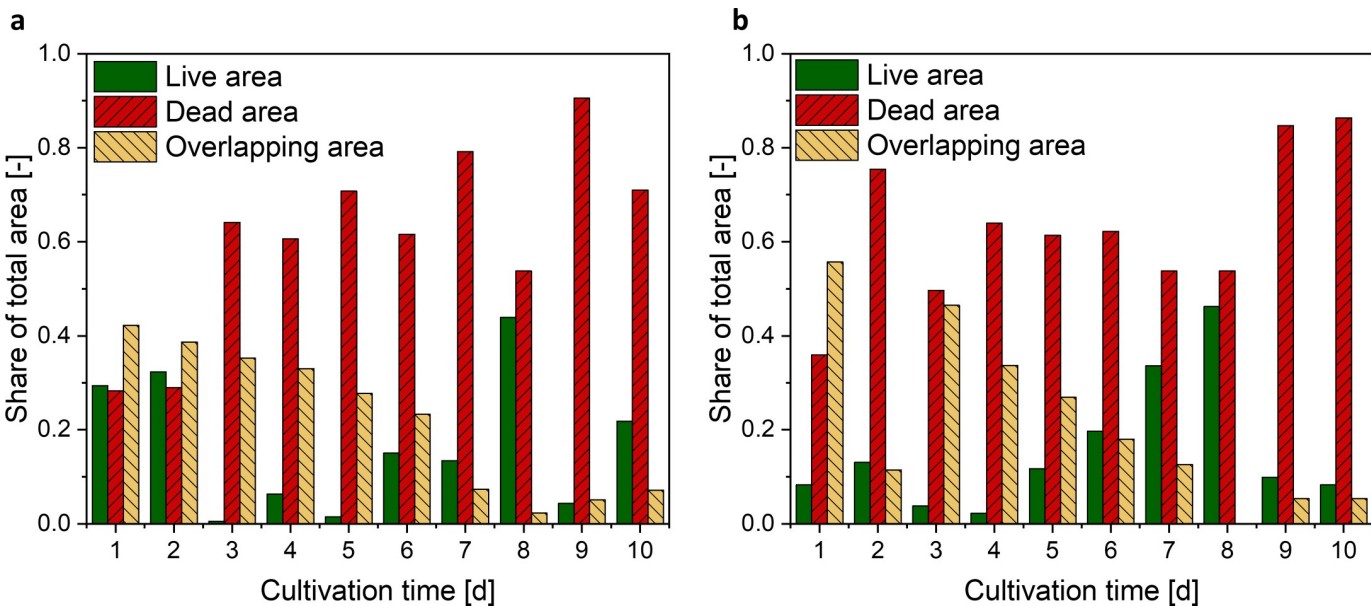

**Fig 5.** Share of live, dead, and overlapping areas of investigated pellets from cultivations (a) <u>without</u> and (b) <u>with</u> glass beads resulting from sliced and stained pellets and CLSM measurements.

cultivated without (**Fig 6A**) and with glass beads (**Fig 6B**). The pellet's core is identified via FSC and SSC signal saturation. Pellets which were cultivated using glass beads clearly display lower overall diameters and more compact structures as evident in FSC and SSC signals. Corresponding fluorescence signal profiles from viability staining also display more uniform trajectories in cultivations with glass beads. These observations stem from more uniform morphological characteristics due to glass bead addition.

Data from many individual pellets can be combined to obtain an overview on populational characteristics. The here-presented data are derived from at least 50 pellets analyzed per sample. Analysis of morphological and physiological responses over cultivation time displays a clear trend as displayed in **Fig 7**: Pellet diameter (**Fig 7A**) is reduced in cultivations with glass particles, while pellet compactness (Eq (5)), **Fig 7B**) is slightly increased. The use of fluorescent staining indicates increased the viability factor in combination with glass beads (Eqs (6–8)) (**Fig 7C**). The autofluorescence factor (Eq (9)) is also higher in these cultivations (**Fig 7D**), which is likely an indicator for enhanced productivity as supported by measurements of rebeccamycin concentration.

## Interlinks between morphology, viability and productivity

All of the analytical methods employed are well aligned when depicting the general trend of the experiments. The here presented data from microscopy and flow cytometry displays lower pellet sizes and more circular shapes. Further analysis on the physiological impact of glass bead supplementation suggests higher viability and autofluorescence in these cultures as confirmed by CLSM and flow cytometry. **Table 1** provides an overview on results obtained via different analytical methods in the scope of this study.

Statistical ANOVA was applied for every day of cultivation to compare different parameters with and without glass bead addition. The statistical analysis of the results from this study showed a statistically significant difference between cultivations with and without glass bead addition with only rare exceptions (see **S1 Table**). Morphological parameters were statistically

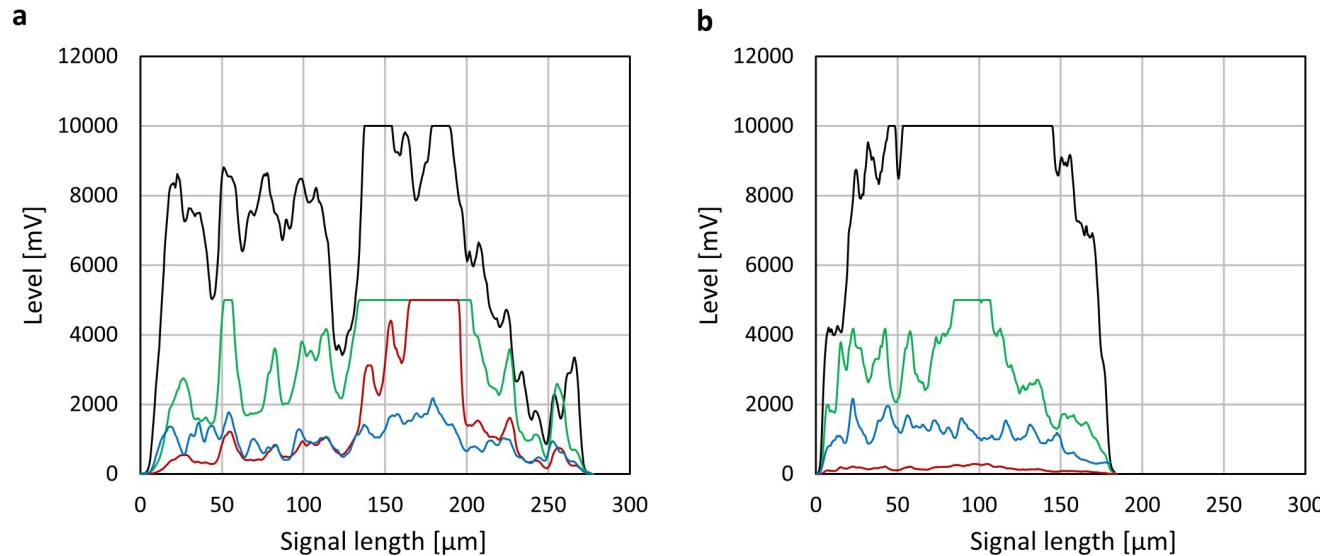

**Fig 6. Exemplary signal profiles of flow cytometry measurements including FSC (black), SSC (blue), green fluorescence (green) and red fluorescence (red) data.** Cultivation (a) <u>without</u> and (b) <u>with</u> glass beads. The pellet's core is identified via FSC and SSC signal saturation. "Length of SSC signal" (Eq (5)) is based on the signal length at half of the maximum of the signal level of the signal profile. To compensate for increased signals at high pellet sizes, "Length of SSC signal" is divided by "Signal length" which represents the overall diameter of the pellet as displayed on the X-axis.

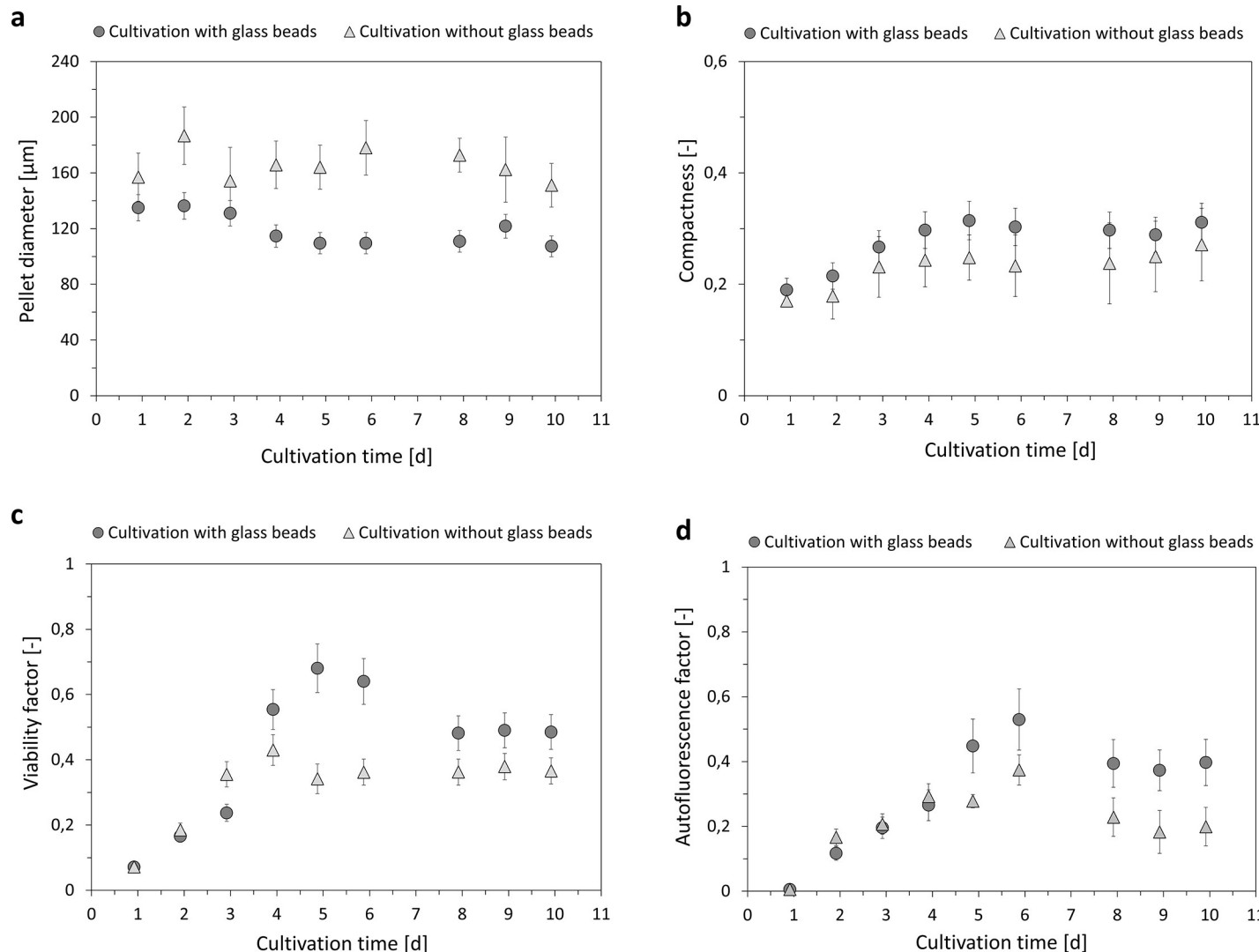

**Fig 7.** Comparison of results from analysis of flow cytometry between cultivations <u>with</u> (circle) and <u>without</u> glass beads (triangle): (a) Pellet diameter derived from signal length, (b) compactness (Eq (5)), (c) viability factor (Eq (8)) and (d) autofluorescence factor (Eq (9)) over cultivation time.

significantly different as well as viability and autofluorescence factor from flow cytometry. The live ratio of pellets gained from CLSM measurements were not tested statistically due to only one pellet being investigated per sample point.

## Discussion

The here presented study strives to provide a complete picture of rebeccamycin productivity and morphology in *L. aerocolonigenes* under glass bead addition. Cell dry weight and substrate concentrations show similar growth phases without and with glass beads (**Fig 2B**), proposing no impact on growth kinetics. The similar cell dry weight concentrations eliminate the possibility of a biomass dependent increased production. This is supported by the biomass-specific rebeccamycin productivity $q_P$, which is increased for the glass bead supplemented cultivation (**Fig 2C**). Significant differences can especially be observed between 3–6 days of cultivation. Macro-morphological analysis of pellet biomass showed smaller and more circular pellets after

**Table 1. Overview on morphological and physiological aspects obtained from pellet analysis via different analytical methods.** Given values are mean values with their respective standard deviations.

| Morphological aspects | Pellet size | | Pellet shape | | |
|---|---|---|---|---|---|
| | *Microscopy* | *Flow cytometry* | *Flow cytometry* | *Microscopy* | *Microscopy* |
| | Mean area-equivalent-spherical-diameter $d_a$ [μm] *Eq (2)*, *Fig 3* | Signal length [μm] *Fig 6* | Compactness [-] *Eq (5)*, *Fig 7* | Circularity [–] *Eq (3)*, *Fig 4* | Ratio minimum/maximum Feret diameter [–] *Fig 4* |
| Without glass beads | 172 ± 61 | 170 ± 64 | 0.22 ± 0.03 | 0.51 ± 0.09 | 0.50 ± 0.07 |
| With glass beads | 106 ± 47 | 100 ± 45 | 0.28 ± 0.04 | 0.64 ± 0.09 | 0.58 ± 0.07 |
| Physiological aspects | Viability | | Autofluorescence, productivity | | |
| | *CLSM* | *Flow cytometry* | *Flow cytometry* | Specific productivity $q_P$ at onset of productive phase [mg g⁻¹ d⁻¹] *Eq (1)*, *Fig 2* | |
| | Live ratio in production phase [–] *Eq (4)*, *Fig 5* | Viability factor [–] *Eq (8)*, *Fig 7* | Autofluorescence factor [–] *Eq (9)*, *Fig 7* | | |
| Without glass beads | 0.10 ± 0.06 | 0.32 ± 0.05 | 0.21 ± 0.10 | 1.35 ± 0.04 | |
| With glass beads | 0.22 ± 0.09 | 0.41 ± 0.11 | 0.32 ± 0.16 | 3.70 ± 0.19 | |

glass bead addition (**Fig 4**). Viability staining of pellet slices showed the trend of an increased pellet viability during the exponential production phase for cultures with glass beads (**Fig 5B**). The morphological and physiological observations were confirmed by flow cytometry (**Figs 6** and **7**). As flow cytometry measurements were conducted using fluorescent staining via FDA, conclusions about metabolic activity of the analyzed pellets can be made. Results imply that the addition of glass particles causes a larger active zone in the pellets and thereby leads to an increased metabolic activity reflected in green fluorescence signals [38–40]. Similar results were obtained for the production of the secondary metabolite lovostatin in *Aspergillus terreus* where larger active, growing zones in pellets occurred to be beneficial for lovastatin formation [41]. Results of the cultivation treated with glass beads also showed smaller pellets but a more regular circular shape. The viability factor (**Fig 7C**) indicates more obvious differences between both approaches, also with an increased viability factor for glass bead treated pellets. Consequently, the here presented data suggests that glass bead addition leads to increased productivity due to enabling a more favorable morphology in the form of smaller and more uniform pellets. Thereby, substrate and oxygen transport into the pellet could be facilitated. As observed for different filamentous fungi, a smaller pellet size is favorable in terms of nutrient and oxygen supply [8, 42–45]. Substrate consumption leads to an increased biomass density and thereby reduces the penetration depths of substrates in the pellets [46–48]. The oxygen transfer into a pellet was investigated in detail for *Aspergillus niger*. Oxygen concentration profiles obtained by microelectrode measurements inside the pellet showed a sufficient oxygen supply for small pellets of around 400 μm in diameter whereas larger pellets of 800 μm diameter or more showed oxygen limitations in the pellet core [8]. Similar observations were made for *A. terreus* [42] and *Penicillium chrysogenum* [43].

A comparison of the morphological and physiological data obtained by different methods shows a general conformity (**Table 1**). Regarding pellet size, the diameters resulting from microscopy and the signal length from flow cytometry are in excellent agreement with 172 ± 61 and 170 ± 64 μm without glass beads and 106 ± 47 and 100 ± 45 μm with glass beads, respectively. Although there are no equivalent pellet shape parameters from microscopy and flow cytometry, all given parameters indicate more defined and more circular pellets with glass bead addition. Hence, this study presents two valid methods for the cell morphological characterization of *L. aerocolonigenes*. An increased viability for glass bead addition is noticeable with both methods, although the numerical values differ. The low number of samples during

viability staining of pellet slices (1 pellet per sample) creates a level of uncertainty regarding reliability of the results. Hence, the flow cytometry was applied to verify the results. With higher numbers of samples (at least 50 pellets per sample) the results are statistically sound and support the first trend of an increased metabolic activity during the production phase. The autofluorescence detected in flow cytometry is higher for the glass bead supplemented cultivation where a connection to an increased productivity is proposed, since rebeccamycin can act as a fluorescent substance itself due to its chemical structure (belonging to the group of indolocarbazoles) [49].

However, the results from this study are partly contrary to previous studies regarding the macro-morphology of *L. aerocolonigenes*. In Schrader et al. [32] pellets were larger with glass bead addition although it had the same effect on rebeccamycin production. In Walisko et al. [29] pellets after glass bead addition were smaller, but became less circular. However, a direct comparison is complicated since all results emerge from different biological approaches that might have had varying cell morphology at the beginning of the cultivation (different pre-cultures). Although the impact on macro-morphology is inconclusive, the addition of glass beads increased the production of rebeccamycin in all studies. Thus, the working hypothesis, that the change in the macro-morphology is not the main reason for the increased rebeccamycin production when adding glass beads is proposed. Although morphology and productivity are often linked for filamentous microorganisms, some cases stating otherwise for certain microorganisms are described in literature. For *Streptomyces clavuligerus* and *Streptomyces virginiae* similar productivities of clavulanic acid and virginiamycin, respectively, were observed for different (macro-) morphologies [50, 51]. The presented results rather suggest further influencing factors like micro-morphology. The influence of the micro-morphological branching rate was described by Wardell et al. [52]. Mutants of *Saccharopolyspora erythraea* with a reduced branching rate and a thereby enhanced hyphal strength produced larger amounts of erythromycin than the original strain. In this case, the biomass growth for all approaches was similar [52], which was also observed for *L. aerocolonigenes* (**Fig 2B**). Further morphological effects on the inner pellet structure, like a higher pellet porosity, that can lead to a similar facilitated oxygen transport into the pellet, to a better oxygen availability and therefore to a better production performance are possible. Hille et al. [8] showed that a sufficient oxygen supply inside the pellet is not only dependent on the pellet size but also on the biomass density of a pellet in the case of *A. niger*. Another conceivable option might be the stimulation of the metabolic activity by mechanical stress of the right magnitude induced by the glass beads.

Nevertheless, the here presented results as well as all glass particle related studies cited confirm that addition of 100 g L$^{-1}$ glass beads with the mean diameter of 969 μm significantly increase rebeccamycin titers in cultivations of *L. aerocolonigenes*. A negligible correlation between pellet size and rebeccamycin productivity is postulated. The change in pellet size after glass bead addition does not seem to be the main factor for an increased productivity. The here applied methods for macro-morphological analysis of the pellets showed very similar results. Both conventional microscopy with subsequent image analysis and flow cytometric analysis seem equally suitable for this purpose. In addition to the viability staining, pellet slicing and imaging with CLSM, the morphological and physiological assessment via flow cytometry is an additional powerful and statistically safe method to quantify the vitality of pellets in a faster and more reliable way than before.

## Supporting information

**S1 Fig. Exemplary flow cytometry scatterplot.** Populations of viable biomass (green), nonviable biomass (red) and background (brown) based on green (FDA positive) and red

fluorescence (PI positive) signals. Total fluorescence signals represent the area underneath the curve with respect to flow cytometry signal profiles as depicted in Fig 6.
(TIF)

**S2 Fig. Normalized number-density-distribution $q0(d_a)$ and volume-density-distribution $q3(d_a)$ of pellets analyzed through microscopic image analysis per day.** $q0(d_a)$ of pellets from cultures (a) without and (b) with glass beads and $q3(d_a)$ of pellets from cultures (c) without and (d) with glass beads.
(TIF)

**S1 Table. Single-factor analysis of variance (ANOVA).** Statistical analysis of study results with $\alpha = 0.05$.
(XLSX)

## Author Contributions

**Conceptualization:** Kathrin Schrinner, Lukas Veiter, Stefan Schmideder, Marcel Schrader.

**Funding acquisition:** Arno Kwade, Heiko Briesen, Christoph Herwig, Rainer Krull.

**Investigation:** Kathrin Schrinner, Lukas Veiter, Stefan Schmideder, Philipp Doppler, Nadine Münch, Kristin Althof.

**Methodology:** Kathrin Schrinner, Lukas Veiter, Stefan Schmideder, Philipp Doppler, Nadine Münch, Kristin Althof.

**Supervision:** Arno Kwade, Heiko Briesen, Christoph Herwig, Rainer Krull.

**Writing – original draft:** Kathrin Schrinner, Lukas Veiter, Stefan Schmideder.

**Writing – review & editing:** Marcel Schrader, Arno Kwade, Heiko Briesen, Christoph Herwig, Rainer Krull.

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
