## [Decision Letter · Decision Letter 0]

1 May 2020

PONE-D-20-08433

Morphological and physiological characterization of filamentous Lentzea aerocolonigenes: Comparison of biopellets by microscopy and flow cytometry

PLOS ONE

Dear Prof. Dr. Krull,

Thank you for submitting your manuscript to PLOS ONE. After careful consideration, we feel that it has merit but does not fully meet PLOS ONE’s publication criteria as it currently stands. Therefore, we invite you to submit a revised version of the manuscript that addresses the points raised during the review process.

We would appreciate receiving your revised manuscript by Jun 15 2020 11:59PM. To enhance the reproducibility of your results, we recommend that if applicable you deposit your laboratory protocols in protocols.io, where a protocol can be assigned its own identifier (DOI) such that it can be cited independently in the future. For instructions see: http://journals.plos.org/plosone/s/submission-guidelines#loc-laboratory-protocols

We look forward to receiving your revised manuscript.

Kind regards,

Leonidas Matsakas

Academic Editor

PLOS ONE

3. Thank you for providing the following Funding Statement: 

"The authors gratefully acknowledge financial support by the German Research Foundation (DFG) in the Priority Programme 1934 DiSPBiotech – Dispersity, structural and phase modifications of proteins and biological agglomerates in biotechnological processes (SPP 1934 DiSPBiotech – 315384307 (HB) and 315457657 (RK/AK)). The work of LV and CH was supported by the Austrian Research Promotion Agency (FFG) (Grant number: 844608) and within the framework of the Competence Center CHASE GmbH, funded by the Austrian Research Promotion Agency (grant number 868615) as part of the COMET program-Competence Centers for Excellent Technologies by BMVIT, BMDW, the Federal Provinces of Upper Austria and Vienna.

We note that one or more of the authors is affiliated with the funding organization, indicating the funder may have had some role in the design, data collection, analysis or preparation of your manuscript for publication; in other words, the funder played an indirect role through the participation of the co-authors.

If the funding organization did not play a role in the study design, data collection and analysis, decision to publish, or preparation of the manuscript and only provided financial support in the form of authors' salaries and/or research materials, please review your statements relating to the author contributions, and ensure you have specifically and accurately indicated the role(s) that these authors had in your study in the Author Contributions section of the online submission form. Please make any necessary amendments directly within this section of the online submission form.  Please also update your Funding Statement to include the following statement: “The funder provided support in the form of salaries for authors [insert relevant initials], but did not have any additional role in the study design, data collection and analysis, decision to publish, or preparation of the manuscript. The specific roles of these authors are articulated in the ‘author contributions’ section.”

If the funding organization did have an additional role, please state and explain that role within your Funding Statement.

Please also provide an updated Competing Interests Statement declaring this commercial affiliation along with any other relevant declarations relating to employment, consultancy, patents, products in development, or marketed products, etc.  

Reviewers' comments:

Reviewer's Responses to Questions

**Comments to the Author**

1. Is the manuscript technically sound, and do the data support the conclusions?

Reviewer #1: Partly

Reviewer #2: Yes

2. Has the statistical analysis been performed appropriately and rigorously? 

Reviewer #1: Yes

Reviewer #2: No

3. Have the authors made all data underlying the findings in their manuscript fully available?

Reviewer #1: No

Reviewer #2: Yes

4. Is the manuscript presented in an intelligible fashion and written in standard English?

Reviewer #1: Yes

Reviewer #2: Yes

5. Review Comments to the Author

Reviewer #1: Manuscript Number PONE-D-20-08433

The present manuscript studied L. aerocolonigenes cultures, comparing two different methods for morphological and physiological characterization.

Generally, the manuscript is well written and structured. However, there are a few aspects that should be improved in order to make the work clearer.

General comments:

Legends should be inserted close to the Figures, in order to make easy the reading of the paper.

Figures are not presented sequentially; Figure 4 is not referred in the text.

Specific comments:

Line 116 – Authors should specify if glucose was separately sterilized;

Line 177 – Authors referred that they have used Syto9 and PI dyes for CLSM analysis; however, they did not explain how these dyes work, as they did - and well done – for the Flow cytometry protocol, in which they explained how PI and FDA work (Line 221).

Line 223 – Please indicate the number of events/s, threshold and compensation settings for FC analysis;

Line 276 – “The transition from growth phase to stationary phase occurs between day 1 and 2 (Fig 2b)”. This is wrong, since the growth phases are identified in natural logarithmic (Ln) biomass versus time plots, not in biomass versus time plots, as Fig. 2b) shows. Please correct the sentence.

Line 338 – “Due to the high manual effort of pellet slicing only one pellet per sample point was examined.” Do Authors think that this procedure is statistically relevant? Please comment.

-It would be interesting to see the cytograms with FDA/PI results

Figures:

Figure 2 a) –Some experimental points show an error > 20%, which is unacceptable; The same for the biomass curve depicted in Figure 2 b)

More experimental points should be presented during the first 24 h, in order to have a more accurate biomass profile.

Figures 3) and b) should present error bars.

Reviewer #2: I have read throughout the manuscript; it’s a valuable and comprehensive research showing metabolic activity in macro morphology of filamentous bacteria producing rebeccamycin. However, in several instances, the presentation and the discussion of the achieved results and the comparison with the relevant literature are rather limited.

Minor Comments: _

1. Provide a suitable reference for sentence mentioned in line no 405 to 406 for metabolic activity.

2. Is there any evidence that rebeccamycin has fluorescence properties (Line no 427-429).

3. Any suitable experiment/reference showing that the smaller and uniform morphology support more substrate uptake and oxygen into pellet. However, results are contradictory, where all glucose are utilized in 2 days under both condition (With glass beads and without glass beads. Explain this.

4. Did you optimize the substrate concentration during experiment (>3.5 g/L).

5. Why authors used only 100 g/L of beads or with same diameter, did they check if more or less quantity of beads have different observation than the current study?

6. Please provide statistical analysis of data.

6. PLOS authors have the option to publish the peer review history of their article (what does this mean?). If published, this will include your full peer review and any attached files.

Reviewer #1: No

Reviewer #2: Yes: ALOK PATEL

---

## [Author Response · Author response to Decision Letter 0]

14 May 2020

Thank you very much for your letter with the reviewer assessment. In the name of all authors I would like to express our gratitude for the fast processing of our manuscript. We appreciate the comments on our paper. The suggestions given were processed. The revised manuscript was resubmitted as instructed. 

We include in our revised manuscript:

- A rebuttal letter that responds to each point raised by the academic editor and reviewer(s). This letter was uploaded as separate file and labeled 'Response to Reviewers'.

- A marked-up copy of your manuscript that highlights changes made to the original version. This file was uploaded as separate file and labeled 'Revised Manuscript with Track Changes'.

- An unmarked version of your revised paper without tracked changes. This file was uploaded as separate file and labeled 'Manuscript'.

Kind regards,

Rainer Krull (for all authors)

---

## [Decision Letter · Decision Letter 1]

20 May 2020

Morphological and physiological characterization of filamentous Lentzea aerocolonigenes: Comparison of biopellets by microscopy and flow cytometry

PONE-D-20-08433R1

Dear Dr. Krull,

We are pleased to inform you that your manuscript has been judged scientifically suitable for publication and will be formally accepted for publication once it complies with all outstanding technical requirements.

With kind regards,

Leonidas Matsakas

Academic Editor

PLOS ONE

Additional Editor Comments (optional):

Reviewers' comments:

Reviewer's Responses to Questions

**Comments to the Author**

1. If the authors have adequately addressed your comments raised in a previous round of review and you feel that this manuscript is now acceptable for publication, you may indicate that here to bypass the “Comments to the Author” section, enter your conflict of interest statement in the “Confidential to Editor” section, and submit your "Accept" recommendation.

Reviewer #2: All comments have been addressed

2. Is the manuscript technically sound, and do the data support the conclusions?

Reviewer #2: Yes

3. Has the statistical analysis been performed appropriately and rigorously? 

Reviewer #2: Yes

4. Have the authors made all data underlying the findings in their manuscript fully available?

Reviewer #2: Yes

5. Is the manuscript presented in an intelligible fashion and written in standard English?

Reviewer #2: Yes

6. Review Comments to the Author

Reviewer #2: Authors responded to all queries raised by reviewer satisfactorily, so I recommend acceptance for this article.

7. PLOS authors have the option to publish the peer review history of their article (what does this mean?). If published, this will include your full peer review and any attached files.

Reviewer #2: Yes: ALOK PATEL

---

## [Editor Report · Acceptance letter]

26 May 2020

PONE-D-20-08433R1 

Morphological and physiological characterization of filamentous *Lentzea aerocolonigenes*: Comparison of biopellets by microscopy and flow cytometry 

Dear Dr. Krull:

I am pleased to inform you that your manuscript has been deemed suitable for publication in PLOS ONE. Congratulations! Your manuscript is now with our production department. 

With kind regards,

on behalf of

Dr. Leonidas Matsakas 

Academic Editor

PLOS ONE